# Unveiling the Role of CCL3: A Driver of CIPN in Colon Cancer Patients?

**DOI:** 10.3390/biomedicines13102512

**Published:** 2025-10-15

**Authors:** Irene Luzac, Cynthia Rosa Regalado, Mihály Balogh

**Affiliations:** Department of Molecular Pharmacology, Groningen Research Institute of Pharmacy, Faculty of Science and Engineering, University of Groningen, 9712 CP Groningen, The Netherlands

**Keywords:** cancer neuroscience, CCL3, colon cancer, chemotherapy-induced neuropathic pain, CIPN

## Abstract

Cancer neuroscience is an emerging field revealing how malignancies interact with the nervous system to shape disease progression and symptom burden. In colorectal cancer (CRC), increasing evidence suggests a direct interplay between tumor cells and peripheral sensory neurons, contributing not only to cancer progression but also to chemotherapy-induced side effects such as peripheral neuropathy. Chemokines, particularly CCL3, appear to be key players in this bidirectional communication. This literature review aims to critically examine the role of CCL3 in CRC and chemotherapy-induced peripheral neuropathy (CIPN), with a focus on identifying potential mechanistic overlaps. Specifically, we evaluate whether CCL3 may serve as a molecular link between cancer progression and the development of neuropathic pain. In CRC, CCL3 is frequently upregulated, promoting tumor proliferation, invasion, and immune remodeling through CCR5- and MAPK-dependent pathways. Elevated CCL3 expression correlates with advanced stage, nerve infiltration, and worse prognosis, while select studies suggest it may also enhance antitumor immunity via dendritic cell recruitment. In parallel, CCL3 is also upregulated in the nervous system during CIPN, where it contributes to chronic pain through activation of glial cells, sensitization of nociceptive pathways (e.g., TRPV1, P2X7), and desensitization of opioid receptors. Notably, MAPK signaling is a shared downstream pathway in both contexts, suggesting a potential mechanistic bridge between tumor biology and neurotoxicity. In conclusion, CCL3 emerges as a central molecule at the intersection of CRC and CIPN. Understanding its context-dependent roles may offer new opportunities for risk prediction, biomarker development, and therapeutic intervention—contributing to the broader goals of cancer neuroscience in improving both oncologic and neurologic outcomes.

## 1. Introduction

In the United States, cancer is the second leading cause of death. According to recent data from the American National Cancer Institute (NCI), colorectal cancer (CRC) is projected to account for 7.6% of all new cancer cases in the U.S. in 2025, making it the fourth most diagnosed cancer. Additionally, CRC is expected to cause 8.6% of all cancer-related deaths [1]. There are a lot of different factors that cause an increased risk of CRC, among these are environmental and genetic factors. For those suffering from cancer, the adverse effects of available treatments can be severe, that is, the patients’ quality of life might be significantly hampered. Studies have shown that a large number of patients that undergo chemotherapy (still a cornerstone of CRC treatment), have to deal with terrible side effects. Chemotherapy-induced neuropathic pain (CIPN) is the most common dose-limiting adverse effect that occurs during treatment [2]. Briefly, CIPN is caused by damage to the peripheral nerves, which leads to debilitating symptoms such as heightened sensitivity to hot and cold, pain and numbness [3]. Because of the severe symptoms, the development of CIPN is the most frequent factor leading to the alteration or termination of cancer treatment and thus affecting survival rates [4]. Data from a cohort study performed in 2016 suggest that CIPN prevalence is 22% after treatment cycle 3 and 87% after treatment cycle 6 [5]. Another study found that 12% of CRC survivors reported CIPN lasting longer than three years post-treatment [6]. Given the necessity of chemotherapy and the severity of CIPN, more research on both cancer and CIPN is needed to develop alternative treatments, improved screening methods, or better prevention strategies.

The pathology of CIPN include the alteration of microtubule-based axonal transport, altered ionic homeostasis, or damage to mitochondrial function [7]. However, there is now a growing attention on neuroimmune interactions in the development of different neuropathies, including CIPN. Most interestingly, numerous studies imply that chemokines have a large role in the development of both colon cancer and CIPN [8].

### 1.1. Definition and Function of Chemokines

Chemokines are a type of cytokine (chemotactic cytokine to be precise) and play an important role in the immune system [9]. Their function is to fight infections, and they are involved in various inflammatory conditions, and other diseases [10]. They exert their functions by signaling through cell surface G protein-coupled heptahelical chemokine receptors [9] and inducing the migration of particular types of white blood cells (leukocytes) to the site of infection or damage [10]. The release and expression of the chemokines are induced by dendritic cells as an immune response [11].

Chemokines are also involved in lymphoid tissue development [7] and angiogenesis [12]. Other biological processes where chemokines have an important role are proliferation, survival, differentiation, cytokine production, degranulation and respiratory burst [13]. It is also interesting that nonleukocytic cancer cells can develop receptors to interact with chemokines, which can result in local invasion, spreading to lymphatic nodes and the metastatic seeding of distant tissues [14,15]. Thus, chemokines significantly influence tumor growth and metastasis.

Post-translational modification, binding to heptahelical ‘atypical’ chemokine receptors and interaction with the extracellular matrix (ECM) are found to affect the chemokines with regard to localization and abundance [9]. Based on the position of the N-terminal cysteine residues, the chemokines are divided into four subfamilies: CXC, CC, CX3C and XC [16]. Several studies show that tumors induce the upregulation of pro-inflammatory proteins, including chemokines. This increase in pro-inflammatory chemokines can induce the development of colon cancer [7]. CCL3 might be particularly interesting in this regard, as it seems that CCL3 expression correlates with the proliferation, invasion, and migration of colorectal cancer cells [17,18], in addition to its feasible role in the development of CIPN [19]. CCL3 exerts its various effects by binding to the immune cell receptors CCR4, CCR1, and CCR5 [20]. This review will focus on a chemokine from the chemokine group involved with inflammation, the CC-ligand (CCL) chemokine CCL3.

### 1.2. CIPN and the Role of Chemokines

As already stated, CIPN is a severe side effect of cancer treatment. Patients treated with oxaliplatin, (a widely used chemotherapeutic agent for CRC), often suffer from CIPN lasting between six months and two years after completing therapy [21]. Chemotherapy induces apoptosis, mitochondrial damage and increases in reactive oxygen species (ROS), triggering various molecular pathways that upregulate pro-inflammatory cytokines and chemokines [19].

In particular, astrocyte activation leads to further production of cytokines and chemokines, including CCL3, which subsequently activates microglia. This process amplifies the release of pro-inflammatory mediators, resulting in neuronal inflammation and fiber degeneration at the synapses [19]. Moreover, chemokine upregulation has been linked to mitochondrial dysfunction [22], which itself is considered a primary driver of neuropathy and neuropathic pain [23,24,25]. However, the specific risk factors underlying intractable, chronic CIPN remain largely unknown.

This review aims to clarify the role of CCL3 in both colon cancer and CIPN. This aim is partially based on a recent study, in which CCL3 was shown to be upregulated in colon tumor bearing mice (with no chemotherapeutic treatment). Interestingly, this increase was concomitant with systemic neuronal dysfunction, closely resembling the symptoms of CIPN [26]. These observations raise the possibility that CCL3 is not only a driver of tumor progression but may also predispose patients to intractable neuropathic complications. We therefore hypothesize that CCL3 functions as a molecular bridge linking tumor biology with neurotoxicity. In this review, we evaluate current evidence to explore the dual role of CCL3 at the interface of cancer progression and chemotherapy-induced neuronal damage.

## 2. CCL3 and Cancer

### 2.1. Positive Correlation of CCL3 Increase with Tumor Growth

Six key recent studies demonstrate how elevated CCL3 drives cancer cell proliferation, invasion, and metastatic potential. Ma et al. has shown that the upregulation of CCL3 (and CCR5) promotes the proliferation, invasion, and migration of colorectal cancer cells via the TRAF6/NF-κB signaling [17]. This study suggested that ligand binding to CCR5 induces GTPase-activating proteins, which then activates protein kinase C (PKC). Subsequently, PKC activates the p65 subunit of a nuclear factor κB (NF-κB) and phosphatidylinol-3-kinase (PI3K) signaling pathway. Activation of the PI3K signaling pathway enables the TNF receptor-associated 6 (TRAF-6) to bind to the TNF superfamily and increase the level of chemokines. TRAF-6 also activates NF-κB in the nucleus via cytokine MyD88, CD4, and AP-1, and this activation leads to biological effects that play a role in the process of inflammation and tumor invasion. NF-κB also inhibits apoptosis and promotes the progression of CRC. Further correlation between CCL3/CCR5 and the clinical indicators was based on the characteristics of clinical data of patients with CRC. The protein array for chemokines, Western blotting, and immunohistochemistry consistently showed that CCL3 and CCR5 levels are significantly elevated in both primary and metastatic tissues, with their expression correlating positively with TNM stage (Tumor-Node-Metastasis; including stages I and III) and nerve invasion. Inhibition of CCL3 signaling (via CCR5 blockade) led to significant decreases in TRAF6 and NF-κB activation and a corresponding drop in tumor cell proliferation. Conversely, CCL3 overexpression enhanced TRAF6 and NF-κB levels and drove increased tumor growth. These findings indicate a crucial role of CCL3 in colorectal cancer (Figure 1).

De la Fuente López [18] et al. confirmed Ma et al.’s [17] observations in a clinical cohort (*n* = 48), reporting significantly higher plasma levels of CCL2, CCL3, and CCL4 in CRC patients versus healthy controls. Plasma CCL3 also correlated with vascular endothelial growth factor (VEGF), TNF-α, and certain clinicopathological features, although its tumor-tissue levels showed no association with macrophage markers, TNM stage, differentiation grade, desmoplasia, inflammation score, or lymphatic metastasis. An effect of age on CCL3 was seen only in controls, implying that elevated CCL3 in patients is CRC-driven. Based on these results, the authors proposed chemokines as CRC biomarkers and potential therapeutic targets. Altogether, these data underscore a key role for chemokines in colon cancer development.

Based on this, CCL3 plasma levels might be used as biomarker in the CRC prognosis, but this warrants further research. Overall, the observed correlations between the levels of the chemokines, macrophage molecular markers, and clinical-pathological characteristics suggest that chemokines have an important role in the development of colon cancer.

In accordance with the studies described above, the study of Lewandowska et al. confirmed that CCL3 is overexpressed in colorectal tumor tissue compared to healthy mucosa, with levels rising alongside node (N) stage but showing no association with distant metastasis (M) or histopathological grade (G) [27].

Phinney et al. [28] then showed in CT26 and MC38 flank-tumor models (Balb/c and C57BL/6 mice) that the MAPK-activated protein kinase 2 (MK2) pathway controls CCL3 production in both tumor cells and macrophages. Some mice were intratumorally injected with cells exposed to MK inhibitors or chemokines or vehicle control. For the observation they made use of Western blotting for the incubated cells. For the analysis of the tumor samples, they used rt-PCR and confocal microscopy. They revealed that CCL3 also induces MK2 phosphorylation, creating a feedback loop. The addition of the chemokine also induced the production of IL-1β, IL-6, and TNF-α via the activation of the MK2 pathway. This defines the dual role of the chemokines in colon cancer development; by increasing the chemotaxis of the macrophages and by promoting tumor cell proliferation. Based on their results, this study suggests that the MK2 pathway is necessary for colon tumor growth. Altogether, this study showed that the MK2 pathway regulates the production of CCL3, which increases tumor cell proliferation and induces the production of other cytokines. Thus, the MK2 pathway is essential for colon tumor growth.

Most recently, Duggirala et al. [29] reported in 2025 that CCL3 is actively involved in the remodeling of the pancreatic tumor microenvironment. Transcriptomic analyses revealed that pancreatic stromal cells co-cultured with PAK1-modulated cancer cells exhibited elevated CCL3 expression, which was linked to activation of interferon pathways and myofibroblast differentiation. The upregulation of CCL3 appeared to mediate crosstalk between stromal and malignant cells, potentially facilitating immune evasion or chronic inflammation. These findings implicate CCL3 as a pro-tumorigenic mediator that modulates the cytokine milieu and supports stromal reprogramming in pancreatic cancer.

This immunomodulatory role of CCL3 has also been observed in other tumor types. In 2024 a study was performed on the cytokine profiles of pediatric patients by Chen et al. [30]. They found significantly elevated levels of CCL3 in the cerebrospinal fluid (CSF) of children with metastatic medulloblastoma compared to non-metastatic cases. The study utilized multiplex immunoassays to assess inflammatory mediators in CSF samples, revealing CCL3 as one of the most indicative biomarkers associated with metastatic disease progression. Although the study did not directly address how CCL3 affects immune regulation, the chemokine’s elevation in metastatic cases suggests its involvement in tumor dissemination and microenvironmental remodeling. These findings support the potential of CCL3 as a prognostic or diagnostic biomarker for CNS tumor metastasis.

Taken together, these six lines of evidence establish CCL3 as a central driver of colorectal cancer progression. Ma et al. demonstrated that CCL3/CCR5 signaling activates TRAF6/NF-κB to promote tumor cell proliferation, invasion, and survival [17]. De la Fuente López et al. confirmed elevated circulating CCL3 (and related chemokines) in CRC patients and proposed its utility as a noninvasive prognostic biomarker [18]. Lewandowska et al. showed tissue-level CCL3 expression rises with lymph node involvement, underscoring its link to metastatic potential [27], while Phinney et al. revealed an MK2-dependent feedback loop in which CCL3 not only fuels tumor growth but also amplifies inflammatory cytokine release [28]. Additionally, Duggirala et al. [29] demonstrated that CCL3 expression in pancreatic stromal cells supports tumor-promoting stromal remodeling and cytokine signaling, suggesting its broader role in shaping pro-tumorigenic microenvironments beyond colorectal cancer. Likewise, Chen et al. [30] identified CCL3 as a promising biomarker of metastatic potential in pediatric medulloblastoma, highlighting its relevance in central nervous system tumors and further supporting its role in cancer progression.

#### 2.1.1. CCL3/CCR5-Driven Crosstalk in the Tumor Microenvironment

In the studies that follow, we review how CCL3/CCR5 signaling influences key stromal and immune cell populations to drive colitis-associated carcinogenesis, tumor–stroma interactions, and metastatic progression.

The study by Sasaki et al., demonstrated that mice deficient in CCL3 or CCR5 exhibited reduced tumor formation following azoxymethane (AOM) and dextran sulfate sodium (DSS) induction [31]. DSS is used to mimic the pathologic changes in human ulcerative colitis [32] and AOM is known to promote the carcinogenesis process [33]. This study utilized CCR5-, CCR1-, and CCL3-deficient wild-type Balb/c mice that were injected intraperitoneally with azoxymethane (AOM) and for 5 days dextran sulfate sodium (DSS) was dispensed via the drinking water. A murine adenocarcinoma cell line, colon 26, was suspended and then administered to the mice. After 14 days, the tumors were removed and prepared for immunohistochemical analysis, qRT-PCR, or flow cytometry. Sasaki et al. found that the expression levels of CCL3, CCR1, and CCR5 were augmented in the colons of mice throughout the whole carcinogen process. Additionally, CCR1-deficient mice developed significantly bigger and more tumors than CCL3-/CCR5-deficient mice. They also revealed that collagen deposition and the amount of intratumoral type I collagen-positive cells were increased in CCR1-deficient mice, but not in CCL3-/CCR5-deficient mice. In addition, it was observed that fibroblasts accumulated after the tumors started to appear in the colon. Because fibroblasts are known to produce various growth factors, the expression of these factors was also examined starting the day the fibroblasts accumulated. According to the authors heparin-binding epidermal growth factor (HB-EGF) expression was reduced in CCL3/CCR5-deficient mice. And their double-color immunofluorescence showed that HB-EGF was expressed by type I collagen-positive cells. Additionally, CCL3 induced the upregulation of this growth factor together with TNF-alpha. This suggests that CCL3 can sustain HB-EGF expression in fibroblasts that are accumulated in the colon tumor. The reduction in the tumor formation in CCL3/CCR5 deficient mice co-occurred with the reduction in AOM/DSS-induced type I collagen-positive fibroblast accumulation. This was associated with an inhibition of the expression of heparin-binding epidermal growth factor (HB-EGF). In vitro, CCL3 induces the proliferation of fibroblasts and the expression of HB-EGF. Moreover, the blockade of CCR5 resulted in a reduction in tumor formation together with the inhibition of fibroblast accumulation and HB-EGF expression. Based on these findings, Sasaki et al. concluded that CCL3-CCR5- mediated fibroblast activity, in addition to leukocyte infiltration, is crucial for colitis-associated carcinogenesis. Comparable roles of fibroblast-derived signaling in tumor growth have also been reported in a colitis-associated CRC model induced by AOM/DSS [34] and in a lung metastasis model induced by intravenous injection of murine renal carcinoma cells [35].

Based on the prior discussed study [31], Tanabe et al. performed a similar study in 2016 [36]. They also focused on the colitis-associated cancer. Their main focus of investigation was on cancer-associated fibroblasts (CAFs) that accumulate at tumor site through the interaction between CCL3 and CCR5. They investigated the effect of maraviroc, a CCR5- antagonist, on tumor growth by focusing on the CAFs. The tumor arose from the orthotopic injection of mouse or human colon cancer cell lines that were orthotopically injected into the cecal wall. They used male and female BALB/c mice. They found that maraviroc reduced the tumor size approximately by half, yet it did not trigger necrosis or apoptosis in those cells. This indicates that maraviroc works via other cell types rather than on the CRC cells themselves. Immunohistology analysis revealed that maraviroc reduced type I collagen- and α-SMA-positive fibroblasts. The SMA-positive cells serve as an indicator of a distinct type of activated fibrogenic cell, the myofibroblast, which plays a significant role in the process of tissue fibrogenesis [37]. This is very important for tumor growth since fibroblasts are major cellular components of the tumor microenvironment, they can provide the tumor with growth factors, and pro-inflammatory tumor-promoting mediators, and induce neovascularization [38].

The researchers also confirmed the increased intratumoral expression of CCL3, CCL4, and CCR5. Double-color immunofluorescence revealed that CCR5 was mainly expressed in SMA positive cells. Flow cytometric analysis found that CCL3 was expressed in the Ly6G-positive granulocytes and F4/80-positive macrophages. Additionally, in vitro exposure of CCL3 to SMA-positive cells binding to CCR5 resulted in enhanced cell migration and an increase in EGF expression, an essential growth factor for CRC cells, and was inhibited by maraviroc. Summarized, Tanabe et al. found that CCR5 was mainly expressed in SMA positive cells and that CCL3 was expressed in the Ly6G-positive granulocytes and F4/80-positive macrophages. The study also showed that SMA positive cells play a significant role in the process of tissue fibrogenesis and are a major cellular component of the tumor microenvironment, providing the tumor with growth factors and inducing neovascularization.

While both Sasaki et al. [31] and Tanabe et al. [36] investigated CCR5 inhibition in colitis-associated CRC models, their studies emphasize distinct cellular mechanisms—immune modulation versus stromal remodeling—underscoring the multifaceted role of the CCL3/CCR5 axis in tumor development.

Similarly, a clinical study carried out by Nishikawa et al. revealed that bone marrow-derived mesenchymal stem cells promote CRC progression via CCR5 activation [39]. In this study, Nishikawa et al. proposed a different point of view on tumor development focusing on the bone marrow (BM). During the development of tumors, mesenchymal stem cells (MSCs) are attracted from the BM to the stroma, where they play an important role in the tumor microenvironment. Their function involves the secretion of growth factors, cytokines, and chemokines. The study revealed that CCR5 was highly expressed by MSCs. Interestingly, the expression of CCR5 in primary CRC seem to be correlated with the prognosis of CRC patients. In particular, patients with stage III/IV CCR5-high CRCs had a considerably worse prognosis compared to those with CCR5-low CRCs. A similar prognosis was found for patients with a high preoperative serum level of CCL3.

Interestingly, Nishikawa and colleagues found that CCL3-CCR5 axis functions between MSCs and CRC cells and this leads to tumor growth in vivo. Co-administration of CRC with MSCs resulted in significantly accelerated tumor growth in mice compared to mice that were injected with only CRC cells. This has driven the authors to investigate the role of CCR5 in CRC progression and found the Erk, PI3K, AKT intracellular pathways to be phosphorylated in the HCT116 cells after CCL5 treatment (HCT116-CCR5). Furthermore, they found that the CCR5 axis caused directional migration in the HCT116-CCR5 cells. The injection of MSCs/CRC mixture not only accelerated the tumor growth in mice, the tumors were also significantly larger. Some of those tumor cells contained SMA-positive cells. Upon treatment with maraviroc, the size of HCT116-CCR5 + MSCs tumors significantly diminished while the size of the HCT116-EV(vehicle) + MSCs tumors remained unchanged. This indicates that CCR5 signaling is very important for the interaction between CRC cells and MSCs. Moreover, they reported that high levels of CCL3 and increased CCR5 expression were associated with a poorer prognosis, particularly in later disease stages.

A different study from 2021 by Pervaiz et al. [40] explored the therapeutic potential of CCR5 inhibition in colorectal cancer (CRC) with liver metastasis, using both siRNAs and the CCR5 antagonist maraviroc. The study demonstrated that targeting CCR5 significantly suppressed CRC cell proliferation in vitro and reduced metastasis in vivo. They found that the inhibition of CCR5 by siRNA and maraviroc, suppresses the proliferation of the CRC cells in vitro. Maraviroc exhibited a direct and dose-dependent anti-proliferative effect due to its effective occupation of the CCR5 structures upon exposure, resulting in receptor saturation that reduced tumor proliferation. On the other hand, siRNA displayed a delayed anti-proliferative effect as it only resulted in reduced expression of the receptor after the existing CCR5 receptors had been degraded.

Additionally, they revealed during the in vivo investigation that activation of CCR5 is required for CRC liver metastasis and that the blockage of the receptor, as presumed, suppresses the metastasis. While in the control group, continuous tumor growth was observed, gemcitabine (nucleoside analog, chemotherapeutic agent) induced moderate inhibition of the tumor [41]. Maraviroc showed significant suppression of the tumor growth, witnessed as a complete remission of the tumor in 66% of the rats.

Furthermore, the authors found that CCR5 targeting suppresses colony formation and the migration of CRC cells. This aligns well with their earlier findings (2015) by Pervaiz et al. who revealed that targeting CCR5 by maraviroc induces arrest in G0/G1 phase of the cell cycle in CRC cells [42]. Interestingly, Pervaiz and colleagues have also shown that the level of CCL3 in stage I patients was similar to the control group. While the level in the serum of stage II patients was reduced, followed by a return to normal levels in the serum of stage III and IV patients. Of note, circulating levels may not represent the actual levels at the tumor sites. Correspondingly, CCL3 expression in primary human CRC tissue generally increased with advancing tumor stage. Summarized, their study revealed the importance of CCR5 and its ligands in CRC development and colorectal liver metastasis via proliferation, migration, colony formation and alterations in cell-cycle signaling cascades. However, as also stated in the study, CCR5 interacts with multiple ligands and various signaling cascades to play an important role in metabolic and proliferative events. So, the effects of CCR5 cannot be directly assigned to CCL3 alone and should be further investigated.

#### 2.1.2. The Role of p53 in CCL3 Regulation

To further unveil the regulatory mechanisms underlying CCL3 expression in CRC, it is also important to consider the influence of upstream genetic factors, such as p53.

The p53 protein is a tumor suppressor that prevents the development of cancer by monitoring the health of a cell’s DNA and inducing pathways that promote DNA repair or cell death in response to DNA damage or stress. Mutations in the TP53 gene, which encodes p53, can impair its ability to suppress tumor growth and lead to the development of cancer [43].

Connecting to the study of Pathak et al. focused on the p53 status of human cancer cells in relation to the CCL3 expression after SN38 and radiation treatment [44]. For this study, human CRC (HCT116 with wild-type, heterozygous, and functionally null p53) cells were treated by radiation and SN38. SN38 is an antineoplastic drug, and functions as a topoisomerase I inhibitor, an enzyme that is crucial for DNA replication and cell division [45]. To test the viability of the cells, an increasing dose (2 Gy–10 Gy) of radiation was used. 2 Gy was used as an ID_50_ (maximal radiation dose to cause half of the inhibition of the cells) value for the experiments. They found that both treatments resulted in upregulation in most of the miRNA in HCT116^p53+/+^ cells, associated with the activation of PI3K-AKT and Wnt signaling pathways. MiRNA is an epigenetic regulator and is involved in cell proliferation, differentiation, and apoptosis. This regulator works via gene expression by translational repression or cleavage of the mRNA targets and has the ability to control p53 [46,47,48]. An increase in CCL3 and VEGF was found after radiation and SN38 treatment. While radiation increased the CCL3 level in all the cell lines, the upregulation induced by SN38 was only found in HCT116 ^p53+/+^ and HCT116^P53+/−^ cells. In HCT116^P53−/−^ cells CCL3 seemed to be down-regulated. Hence, the upregulation of CCL3 is correlated with the p53 status in the cells. So, this study demonstrated that the expression of the cyto- and chemokines CCL3 and VEGF are largely dependent on the p53 status of cancer cells, with an increase in CCL3 and VEGF observed in HCT116^p53+/+^ and HCT116^P53+/−^ cells, but not in HCT116^P53−/−^ cells.

#### 2.1.3. Chemokine-Chemokine Receptor Axis as a Treatment Option for Cancer

To leverage CCL3’s elevated expression in colorectal tumors, researchers have begun testing whether redirecting immune cells via the CCL3–CCR axis can improve anticancer efficacy. In this context, Zou et al. confirmed that CCL3 is overexpressed in tumor tissue compared to adjacent healthy mucosa and then explored using this for therapy [49]. They showed that engaging the chemokine–chemokine receptor axis enhances CIK cell migration toward colorectal tumor sites, although CIK cell levels did not correlate with the tumors’ chemokine expression profile. CIK cells are generated by coculturing human peripheral blood mononuclear cells with various cytokines, producing a population with both T-lymphocyte and NK-cell activity capable of killing chemo-resistant tumor cells. In this study, elevated CCL3 levels were exploited by increasing the matching chemokine receptor on CIK cells to enhance their tumor-targeted trafficking.

### 2.2. Negative Correlation Between CCL3 and Tumor Growth

Although the majority of studies draw a positive connection between the upregulation of CCL3 and tumor development, some studies found the opposite or indicated that lower levels of CCL3 might also lead to an elevated cancer risk.

Researchers discovered that lower levels of CCL3 were linked to an increased risk of CRC in a large Japanese case–cohort analysis (457 CRC cases, 774 people subcohort subjects, 18-year follow-up) [50]. Blood samples were analyzed by Luminex for various cytokines and chemokines and stratified by time from blood draw to diagnosis (≤5 years vs. >5 years) and tumor location (colon vs. rectum).

They found that decreased blood levels of CCL3 may be linked to an increased risk of developing colorectal cancer in otherwise healthy individuals. This underscores the prospective role of CCL3 in early risk assessment and screening protocols. However, it does not address variations in CCL3 expression across different stages of established colorectal cancer. Therefore, it is hard to draw conclusions regarding the relationship between CCL3 levels and cancer progression or staging based on this data.

Similar links for both colon and rectal cancers were found by Pender et al. [51], revealing that CCL3-knockout mice are resistant to chemically induced colonic inflammation and tumorigenesis. In their population-based case–cohort study, plasma CCL3 levels were inversely associated with colon and rectal cancer risk. Strengths include a large case count, thorough adjustment for confounders, and blood draws at multiple prediagnostic time points to capture all relevant intervals. However, the observed associations were modest, and the investigated correlations (e.g., correlation between circulating levels of CCL3 and the risk of colon cancer development) were not significant after false discovery rate.

The more recent preclinical study by Allen et al. [52] supports the work by Song et al. [50] demonstrating that autologous tumor-derived CCL3 enhances IFNγ production and promotes dendritic cell maturation within tumor-draining lymph nodes (TDLN) in a CCL3-dependent manner. The authors compared the CCL3-secreting CT26 (L3TU) cell-induced colon tumor to wild-type tumors (WTTU) during the priming phase of the antitumor response. WTTU are ‘typical’ cancer cells, not producing CCL3. For the tumor measurements, the mice were injected in the left flank with L3TU alone or a mixture with the WTTU and the tumor was measured twice weekly. The suppressed tumor growth was accompanied by an in vivo inflammatory response. Additionally, CCL3 enhanced CD8^+^ T cell proliferation and differentiation by upregulating dendritic cell capacity, which results in T-cell activation in the tumor-draining lymph nodes (TDLN). Summarized, the data suggest that CCL3 has a direct effect on the immune system in TDLNs, leading to an increase in IFNγ+ NK cells and CD103^+^ DCs, as well as an improvement in Ag-presentation (antigen-presenting cells present antigens to the T-cells) and stimulation capacity of dendritic cells. This in turn results in improved activation of T and B lymphocytes leading to inhibiting the tumor growth due to their immune-responsive behavior.

Allen et al. performed a similar study in 2018 [53] further demonstrating that CCL3 induces tumor rejection and enhances CD8^+^ T-cell infiltration through NK and CD103^+^ DC recruitment via IFNγ. They investigated how CCL3 enhances antitumor immunity in a murine colon microenvironment. Natural killer (NK) cells seem to be a major lymphocyte subtype recruited predominantly to the CCL3-rich tumor site. NK induces IFNγ, CD103^+^ DC accumulation, and other chemokines’ upregulation (CXCL9 and CXCL10). Furthermore, they found that both soluble CCL3 and CCL3-secreting radiated tumor vaccine can suppress the progression of tumors in a spatial-dependent manner. The authors state that their results indicate the importance of NK cells in the CCL3-CD103^+^ DC-CXCL9/10 signaling axis, in the determination of the immunological profile of the tumor’s microenvironment. In this study, Allen and colleagues used the same cell lines; WTTU and cells engineered to secrete CCL3 (L3TU) were injected into the left flank. To test the antitumor efficacy of CCL3, they conducted two immunotherapeutic approaches to treat the CT26 cell induced colon tumors. The first was performed with irradiated L3TU (il3) whole cell tumor vaccine to threaten the WTTU in vivo. This resulted in a suppressed tumor growth. Additionally, the efficacy of recombinant CCL3 (rCCL3) was investigated. Therefore, high-dose bolus of rCCL3 was s.c. administered either intra-tumorally, in the ipsilateral footpad, or in the contralateral footpad, seven days following the WTTU injection. In WTTU mice, the rCCL3 failed to inhibit the tumor growth and in some mice even induced it. The tumors of L3TU mice were smaller compared to L3TU mice without the rCCL3 treatment. Note that direct intra-tumoral injections of either irradiated WTTU or L3TU, failed to significantly slow the WTTU growth in vivo. Overall, cells secreting CCL3 were linked to reduced tumor growth, and treatments like the IL3 vaccine or bolus rCCL3 therapy showed promise in reducing tumor growth.

Further supporting the immune-enhancing role of CCL3, Yuan et al. [54] revealed that DC-targeting chemokines inhibit CRC progression and state that CCL3 exhibits anti-tumor activities in vivo. The correlation analysis of chemokine mRNA expression with the DC markers (ITGAX and CLEC9A) was performed with the cancer genome atlas dataset (TCGA). The in vitro investigation used murine colorectal tumor cell lines (CT26 and MC38) that stably overexpressed different chemokines, including CCL3, established by lentiviral transduction. The effect of the chemokines on cell proliferation was in vitro evaluated with cell counting kit-8 (CCK-8) and colony formation assay. For the in vivo investigation, C57BL/6 and Balb/c mice were used as synergetic subcutaneous tumor models. Immunohistochemistry was used to examine the Ki-67 expression and with flow cytometry, the immune cells in the tumor microenvironment (TME) and lymph nodes were analyzed. Their findings reveal that the expression of CCL3 positively correlates with the DC markers, which indicates that these chemokines have a positive role in the infiltration of DC in the tumoral cells. Interestingly, tumoral overexpression of DC-targeting chemokines did not have a significant effect on the tumor cell proliferation/survival in vitro, while this process was clearly suppressed in vivo. Additionally, they found that CCL3 increased the percentages of CD45^+^ leukocytes in the MC38 tumor cells.

In the study by Yang et al. [55], the authors identified a crucial role for the centrosomal protein TACC2 in modulating chemokine-mediated immune responses in soft tissue sarcoma (STS), together with a key role of CCL3 in the anti-tumor response. For the role of TACC2, its expression was associated with poor prognosis and reduced response to anti-PD-1 immune checkpoint blockade. PD-1 (Programmed cell Death protein 1) is an immune checkpoint receptor on T cells that, when engaged by its ligands, suppresses T cell activity. This mechanism is used by tumors to evade immune surveillance. On the molecular level, TACC2 promoted the transcription of the chemokines CCL3 and CCL4 by preventing nuclear translocation and thereby facilitating CD8^+^ T-cell infiltration into the tumor microenvironment. The mouse models demonstrated that enforced TACC2 expression significantly enhanced the efficacy of PD-1 blockade, resulting in increased tumor suppression and survival. These findings position CCL3 as a downstream effector of TACC2-mediated immune activation, highlighting its possible relevance in augmenting anti-tumor immunity and the effectiveness of immunotherapy in mesenchymal tumors.

Zhang et al. [56] also showed the immune-boosting effect of CCL3 in CD8^+^ T-cell-mediated tumor immunity. In models of lung cancer, genetic knockout of progranulin (PGRN)-a protein known to suppress immune responses and promote tumor progression- resulted in robust upregulation of CCL3 in CD8^+^ T cells, correlating with enhanced infiltration, proliferation, and effector function. Elevated CCL3 expression was directly associated with reduced T-cell exhaustion and increased production of inflammatory cytokines such as IFN-γ. Importantly, CCL3 contributed to the observed synergy between PGRN deletion and anti-PD-1 blockade in enhancing anti-tumor responses. This study positions CCL3 as a key regulator of T-cell functionality in the tumor microenvironment and a potential target for boosting immunotherapeutic efficacy.

So, both Yang et al. [55] and Zhang et al. [56] demonstrate that CCL3 is a critical enhancer of CD8^+^ T-cell-mediated anti-tumor responses and a key determinant of the therapeutic efficacy of PD-1 immune checkpoint blockade, revealing the anti-tumor effects of CCL3.

These epidemiological findings contrast with data from patients with diagnosed colorectal cancer, where CCL3 levels are often elevated and associated with tumor progression and poor prognosis. A possible explanation lies in the biphasic nature of CCL3’s role: it may act protectively in early stages by supporting immune surveillance, while later being co-opted by tumors to promote inflammation, immune suppression, and metastasis. This dual behavior likely depends on the timing of expression, tissue location, and the state of the immune system. The study by Song et al. [50], where lower systemic CCL3 levels correlated with increased CRC risk, suggesting impaired early immune function, is the only epidemiological study to examine pre-diagnostic blood samples in relation to CCL3 and CRC risk, whereas most other investigations have focused on diagnosed patients or tumor tissue, emphasizing progression or prognosis rather than risk. In contrast, in tumor-bearing subjects, elevated CCL3 is often linked to disease progression, particularly through mechanisms like enhanced angiogenesis, recruitment of immunosuppressive cells, and sustained inflammation [17,18,28,31,35,36]. Conversely, some studies have shown that CCL3 can promote antitumor immunity by activating cytotoxic lymphocytes and dendritic cells [49,52,53,54]. These seemingly contradictory results may reflect differences in disease stage, tumor models, or experimental methods, but together they underscore the complex role of CCL3 in cancer biology and its potential as both a biomarker and therapeutic target.

## 3. CCL3 and Neuropathic Pain

### 3.1. CCL3 and CIPN

This section reviews studies linking CCL3 to neuropathic pain, with a focus on chemotherapy-induced peripheral neuropathy (CIPN). Because CIPN-specific data are still scarce, we later also include broader work on CCL3 in neuropathic models with possible implications on CIPN.

Makker et al. characterized immune and neuroinflammatory changes in male C57BL/6J mice after paclitaxel (PTX) or oxaliplatin (OXA) administration [57]. Mice received intraperitoneal injections of PTX or OXA over one week, and tissues were analyzed on day 13. PTX treatment—but not OXA—significantly upregulated CCL3 in the lumbar dorsal root ganglia (DRG) and spinal cord, coinciding with astrocyte activation in the CNS and peripheral neuronal damage. By contrast, neither PTX nor OXA caused detectable leukocyte infiltration into the nervous system, suggesting that neuroinflammation in this model is confined to resident glial and neuronal cells. This statement was also supported by the observation that the depletion of systemic T-reg cells did not affect pain perception in the OXA-treated mice, contradicting the expectation because of their immune-responsive effect. They also tested the neurodamaging effects of the chemotherapeutics on the periphery and CNS. To test peripheral neuronal injury, immunohistochemistry for ATF-3, a marker of cells with damaged peripheral axons, was carried out. Remarkably, only PTX-treated mice showed a significant increase in the neurons expressing ATF-3 in comparison to the control group. To investigate the changes in the CNS, they examined immune-like glial cell changes in the lumbar spinal cord of the treated mice. Again, a significant increase in astrocyte activation was found only in the PTX-treated mice. Interestingly, no significant changes in spinal cord microglia/macrophages were found in the mice with CIPN compared to the control group. Overall, this study proved that leukocyte infiltration into the nervous system of the PTX- and OXA-treated mice was not observed, suggesting that alterations in neuroinflammation are limited to the immunological/glial cells in this in vivo model of CIPN. Additionally, only PTX increased the CCL3 in the PNS and CNS, together with astrocyte activation and neuronal damage. An aspect to take into consideration in this study is the duration of the observation period. The absence of detectable neuronal damage in OXA-treated mice may be related to the timing, as such effects are reported to occur beyond 13 days post-injection. Future research could explore this by including a longer follow-up period.

Next, a case–control study performed by Ochi-ishi et al. investigated the relation between CCL3 and the development of PTX induced neuropathy [58]. This study aimed to examine the involvement of the chemokine CCL3 and the spinal purinoceptor P2X7R in PTX-induced mechanical allodynia. P2X7R is an ionotropic purinergic p2 receptor that is activated by extracellular ATP (adenosine triphosphate). It is a ligand-gated ion channel that opens in response to the binding of ATP and other nucleotides such as ADP and UTP, as well as extracellular bacteria, viruses, and other danger signals. P2X7R is widely expressed in immune cells, such as macrophages and microglia, and plays a crucial role in the activation of the immune response. In terms of pain perception, P2X7R has been implicated in both acute and chronic pain. Activation of P2X7R on sensory neurons can enhance pain signaling, while inhibition of P2X7R can reduce pain sensitivity in animal models of inflammatory and neuropathic pain. P2X7R is also known to be involved in the development of chronic pain conditions [59].

They found that repeated intravenous administration of PTX resulted in mechanical allodynia in rats, accompanied by the upregulation of the mRNAs for CCL3 and its receptor CCR5 in the spinal dorsal horn (SDH) in the CNS with an increase in the number of microglia in the SDH. Interestingly, CCR1 mRNA expression, another CCL3 receptor, remained consistent. Intrathecal administration of a CCL3-neutralizing antibody produced a preventive and reversal effect on PTX-induced neuropathic pain, indicating the role of CCL3 in the maintenance of this phenomenon. Additionally, they demonstrated that the upregulation of the purinoceptor P2X7R receptors (P2X7Rs) was observed in the SDH of PTX-treated rats and contributes to the release of CCL3 from microglia. The selective P2X7R antagonist A438079 was also found to have preventive and reversal effects on PTX-induced neuropathy. The study administered CCL3-neutralizing antibodies and P2X7R antagonists to mice before and after PTX treatment to assess their impact on the development and maintenance of PTX-induced mechanical allodynia. They only measured mechanical allodynia, without assessing thermal thresholds. Results suggest CCL3 plays a crucial role in both the development and maintenance of CIPN, potentially released from microglia via P2X7Rs. Neutralizing CCL3 prevented and reversed neuropathy, indicating its involvement in both mechanical allodynia development and maintenance. Together, these findings define a PTX-induced CCL3–CCR5 neuroimmune circuit underlying CIPN (Figure 2). This circuit may operate as a self-sustaining feedback mechanism: PTX triggers microglial activation and CCL3 upregulation, which in turn drives neuronal injury. The resulting damage may further stimulate glial responses and CCL3 production, perpetuating neuropathic pain.

### 3.2. CCL3 in Other Models or Forms of Neuropathic Pain

The study by Rojewska et al. explored the role of CCL3 in neuropathic pain (NP), focusing on the CNS, and its association with diabetic neuropathy. For this study, streptozotocin (STZ)-induced mouse model of NP was used [60]. STZ causes an elevation in blood glucose levels via damaging the insulin producing pancreatic cells, which is associated with the emergence of long-term hypersensitivity to thermal and mechanical stimuli [61,62,63]. To test the effects of the activated astroglia and microglia cells, they used lipopolysaccharide (LPS) to activate those cells, which then increased the level of CCL3. Their findings revealed that CCL3 and its receptor are predominantly expressed by microglial and astroglia cells; and that the level of the chemokine was increased in the spinal cord in the neuropathic mice by microglial activation and neurons. The found spinal neuronal location of CCR1 and CCR5 explains their role in the nociceptive transmission and explains the pronociceptive effect of CCL3. The authors make the interesting conclusion that neurons are very important in the upregulation of the chemokine, and they derive that from the observed long-lasting effect of the chemokines. However, further mechanistic experiments are required to address this and the exact mechanisms by which neurons contribute to the release of such chemokines. They have also shown that CCL3 and CCR1/5 are co-expressed on the neurons. the development of neuropathy, the protein levels of CCR1 and CCR5, which are located on astrocytes and microglial cells, remained the same. The researchers hypothesize that exposure of these G-protein coupled receptors (GPCRs) to their agonists (CCL3 and CCL9) can result in a decrease in the number of binding sites found on the cell surface, because of the rapid degeneration effects of the agonists on the receptors. It can be inferred that the number of pain receptors remains constant even with increasing levels of pain, contrary to what would be expected. Administration of a CCL3-neutralizing antibody delayed the symptoms of the neurological pain and the injection of a CCR1 antagonist attenuated them, confirming the pronociceptive effect of CCL3.

A recent study carried out by Ruff et al. examined chemokine expression in rat models of STZ-induced diabetic peripheral neuropathy (DPN) and sciatic nerve ligation, focusing on RAP-103, a multi-chemokine receptor antagonist (CCR2/CCR5), for potentiating morphine antinociception and inhibiting neuropathic pain [64]. They reported that CCL3 mRNA was upregulated in the sciatic nerve of STZ-induced DPN rats but not in the spinal cord. RAP-103 significantly reduced CCL3 expression and produced analgesia: it fully reversed mechanical hypersensitivity yet only partially reversed thermal hypersensitivity. Taken together, this study supports Rojewska’s findings [60].

It is necessary to note that diabetic neuropathy and CIPN are of course distinct conditions. Therefore, the findings of studies investigating the role of CCL3 in diabetic neuropathy cannot be directly extrapolated to CIPN. However, these results can still provide some preliminary insight into the potential involvement of CCL3 in CIPN. Overall, based on the findings of these studies the role of CCL3 in directly driving the pain symptoms of neuropathy is confirmed again. The results show that its expression is increased in the spinal cord in mice with neuropathic pain. The administration of CCL3-neutralizing antibodies and a CCR1 antagonist was found to reduce the symptoms of neurological pain, indicating the pronociceptive effect of CCL3. However, the complexity of the intracellular processes in which chemokine GPCRs are involved is extensive, making it difficult to give a complete understanding of the mechanism.

Taken together, Rojewska et al. [60] has shown a correlation between the increased expression of CCL3, microglial activation, and the lowered pain threshold to mechanical and thermal stimuli supporting the molecular mechanism proposed by the study of Ochi-ishi et al. [58]. Note that Ochi-ishi study did find an increase in the CCR5 expression, while the study of Rojewska set al. showed no change in the CCR1/CCR5 receptors levels.

The main focus of the study by Sun et al., performed in 2016, is investigating the role of interleukin-4, the chemokine CCL3 and its receptor CCR5 in a mouse model of neuropathic pain induced by sciatic nerve constriction [65]. They report that astrocytes and microglia produce CCL3 in response to peripheral nerve injury; microglia infiltrate the injury site, become activated, and secrete cytokines/chemokines. Intrathecal injection of a CCL3-neutralizing antibody suppressed p38 MAPK activation and raised pain thresholds, confirming that CCL3 contributes to pain hypersensitivity. Moreover, CCR5 knockout mice showed reduced neuropathic pain behaviors. Plasma collected at multiple time points after nerve constriction had elevated CCL3 compared to controls, consistent with microglial activation. Overall, they hypothesized that nerve injury under low IL-4 conditions leads to microglial activation and p38 MAPK phosphorylation, driving CCL3 release; enhanced CCL3–CCR5 signaling on spinal microglia then amplifies pro-inflammatory cytokine production and sustains neuropathic pain.

Zhang et al. [66] performed transcriptomic analysis in the anterior cingulate cortex (ACC) of CCI (chronic construction injury—a rodent model for neuropathic pain) rats and similarly found that CCL3 mRNA was upregulated seven days after nerve injury, paralleling observations in the spinal cord. This work adds an emotional and affective dimension by implicating ACC changes in neuropathic pain: increased CCL3 and other CCR1 ligands in the ACC suggest that chemokine signaling may also modulate pain-related affect, though the ACC–spinal connection remains complex and poorly understood.

The increase in CCL3 after CCI is also supported by the study of Mert et al. [67] performed in 2021. Collectively, the literature data on diabetic neuropathy or CCI consistently reveals that increased levels of CCL3 are associated with neuropathic pain symptoms.

#### 3.2.1. CCL3-Producing Cells and the Development of Neuropathy

Sun et al. [65] support the mechanism proposed by Rojewska et al. [60] and Ochi-ishi et al. [58], showing that microglial activation is critical for CCL3 upregulation and neuropathy development. This contrasts with Makker et al. [57], who did not observe a correlation between microglial activation and CCL3 expression. Several additional studies confirm the role of microglia as CCL3-producing cells, as discussed below.

Matsushita et al. [68] observed increased CCL3 mRNA expression in the spinal cord following nerve injury, which they attributed to activated CD11b-positive microglia. Their rat model demonstrated that administration of a CCL3-neutralizing antibody could both prevent and reverse tactile and thermal allodynia.

In this study, a spinal nerve injury model was established in rats and the mechanical and thermal allodynia were assessed by the von Frey test and the hot plate test. Similar effects were achieved with CCR1 and CCR5 antagonists, emphasizing the functional significance of these receptors. While technical limitations precluded protein-level verification of CCL3 expression, elevated gene expression was confirmed. They also found CCR5 expression on Iba-1-positive microglia, and observed that CCR5 mRNA levels increased alongside microglial activation in response to partial sciatic nerve ligation. Interestingly, CCR1 and CCR5 were shown to mediate different responses: CCR1 binding caused acute and transient allodynia, while CCR5 induced slower but more sustained effects. The authors propose that these functional differences stem from the receptor expression patterns. For example, CCR5 being upregulated in small glial-like cells and CCR1 in neuron-like cells in deeper spinal regions

Building on these findings, Li et al. [69] conducted a study in 2020 investigating the role of high mobility group box 1 protein (HMGB1) in spared nerve injury (SNI) rats. They showed that HMGB1 regulates CCL3 expression, as an HMGB1-neutralizing antibody suppressed CCL3 levels. Lidocaine treatment reduced neuropathy symptoms by inhibiting HMGB1 and modulating the CCL3/CCR1/CCR5 axis. The authors also confirmed co-expression of CCL3 with the neuronal marker NeuN.

Summed up, this study states that HMGB1 regulates the expression of CCL3 (at least in part) and its receptors and is considered the upstream factor of the CCL3/CCR1/CCR5 signaling pathway.

Anloague et al. [70] reported that CCL3 plays a pivotal role in the pathological interaction between malignant plasma cells and osteocytes in multiple myeloma. Their study showed that CCL3, produced by myeloma cells, induced the release of HMGB1 by osteocytes, which in turn stimulated RANKL expression, a key driver of osteoclast activation and bone resorption. This CCL3-HMGB1 signaling loop contributed to enhanced bone degradation in both murine and ex vivo human models. Elevated CCL3 expression was also observed in bone biopsies from patients with multiple myeloma, where it correlated with increased bone disease and poorer clinical outcomes. These findings implicate CCL3 as a critical mediator of tumor-induced bone remodeling, promoting osteolytic activity through osteocyte reprogramming.

Interestingly, these studies claim the HMGB1 to be very important in the expression and actions of CCL3, but these are the only two studies up to date that mentioned this protein in relation to CCL3. Further, the study of Li et al. [69] confirms the co-expression of CCL3 with NeuN, as the study of Rojewska et al. [60] also revealed.

The next study also expanded on the results of the study of Matsushita et al. [68] and was performed by Kwiatkowski et al. in 2016 [71]. Their findings also support the suggestion that the upregulation of CCL3 is a result of activated microglia. Even more, their data indicate that apart from the microglia, activated astroglia cells are also capable of producing CCL3. The main focus of this study was the investigation of the influence of chronic administration of a CCR5-antagonist (maraviroc) on the nociception and opioid effectiveness during neuropathy induced by CCI to the sciatic nerve. They observed changes in CCR5 expression in the CNS and PNS by using glial cell markers in the spinal cord and DRG. Based on prior literature and their findings, they state that microglial signaling has a crucial role in the development of neuropathy. They propose the following mechanism, as also stated by the study of Sun et al. [65]; microglial cells transform to a reactive form in response to peripheral injury. In their reactive form, they enhance synaptic transmission in the spinal cord via the upregulation of cell-surface receptors and nociceptive factors. Not only do microglia change morphologically, but they also migrate toward the injured region. In addition to the analysis of microglial and astroglial cell cultures, Kwiatkowski and co-workers also performed behavioral tests. The mechanical allodynia was tested with the von Frey test and the thermal hyperalgesia with the cold plate test, which showed that the mice exhibited increased sensitivity to mechanical and thermal stimuli, indicating the presence of these pain conditions. All taken together, this study declares that CCL3 is expressed by microglia and astroglia cells and that its upregulation is potentiated by inflammatory status.

In 2020, Kwiatkowski et al. [72] performed a follow-up study investigating a different CCR5 antagonist, cenicriviroc. The results aligned with their earlier work, but this study introduced two additional mechanisms: (1) heterologous desensitization of opioid receptors by chemokine receptor signaling, and (2) heterodimerization between the µ-opioid receptor (MOR) and CCR5. These mechanisms suggest that CCL3, through CCR5 activation, may interfere with opioid analgesia by inactivating MOR. This offers a novel perspective on how CCL3 may not only contribute to NP but also diminish the efficacy of opioid treatments.

To summarize, multiple studies converge on the conclusion that CCL3 is upregulated in NP and produced primarily by activated microglia, with astroglia also contributing. HMGB1 may act upstream of this signaling, further modulating the CCL3/CCR1/CCR5 pathway. The differential effects of CCR1 versus CCR5 activation, as well as the potential impact on opioid analgesia, underscore the complex role of CCL3 in neuropathy.

#### 3.2.2. CCL3 Influences Opioid and Transient Receptor Potential Vanilloid 1 (TRPV1) Receptor

Kiguchi et al. [73], confirmed the findings of Kwiatkowski et al. [72]. The main focus of this study was to investigate the contribution of CCL3 in the spinal cord to nerve injury-induced NP. This study found that the upregulation of spinal CCL3 and CCR1 participate in NP via central sensitization that is elicited by PNI. Based on literature data showing that CCR1 and CCR5 are expressed by microglia and astrocytes [74,75] and activated glial cells produce CCL3 [76], this study hypothesized that CCL3 participates in NP through activation and cross-talks between the glial cells. The cascades that are activated in these cells are induced by the chemokine receptors that are Gαi-coupled. The detected an increased level of CCL3 first on the third day after nerve ligation. Based on their investigation on a partial sciatic nerve ligation model, they suggest that the upregulation of CCL3 and CCR1 in astrocytes and microglia in the SDH is induced by PNI and binding of CCL3 to its receptor, via the triggering of the MAPK pathway.

Similarly to the study of Sun et al. [65], the study of Kiguchi et al. [73] also provided evidence for a correlation between the upregulation of CCL3 and the activation of the MAPK pathway. The clear involvement of CCL3 in pain perception by neuropathic pain symptoms is further underpinned by various other studies as well.

One of those studies was performed by Zhang et al. [77]. They found that CCR1 was co-expressed with TRPV1 on neurons in the DRG. TRPV1 is involved in the integration of pain signals within nociceptive primary afferents, which are characterized by increased excitability in response to nerve damage. It is now generally accepted that the alteration of TRPV1 can be a major driver of neuropathy [78,79]. In the study, the human embryonic kidney (HEK) 293 cells that were pre-treated with CCL3 showed a 3-fold increase in the sensitivity of TRPV1 and thereby enhanced the response of the DRG neurons to the nociceptor’s agonists. By double-staining of DRG sections, the study revealed that all cells binding to isolectin IB4 and all substance P immunoreactive cells expressed CCR1. These findings suggest that nociceptive neurons express CCR1. Their results also suggest that CCL3 exerts its effect through G-protein-dependent PLC- and PKC-mediated signaling pathways. Based on these findings and the prior literature, the following molecular mechanism was proposed: accumulation of CCL3 activates Gi-coupled chemokine receptors that are present on the neuronal cell surface. This results in the activation of PLC which then sensitizes TRPV1 through PKC phosphorylation. Overall, the findings of this study reveal the role of CCL3 in the sensitization of TRPV1-mediated signaling and therefore in thermal hyperalgesia.

Zhang et al. also found that CCL3 affects pain perception [77]. In agreement with Kwiatkowski et al. [73], they showed that a CCR5-antagonist can increase opioid receptor sensitivity.

Finally, Szabo et al. examined how chemokines can control leukocyte behavior through G protein-coupled receptors and their ability to undergo heterologous desensitization [80]. In this study, they used primary murine thymocytes of male BALB/c mice in addition to human peripheral blood monocytes. In the behavioral tests they applied the cold-water flick test on rats. They hypothesize that chemokines can alter pain perception by cross-desensitization of the opioid GPCRs. Even though this study did not specifically test for the effects of CCL3, they did use another CCR1 ligand and various other chemokines. They found that pre-treatment with a CCR1 agonist (RANTES/CCL5) followed by the administration of an opioid results in a decrease in the pain thresholds. They suggest that the activation of pro-inflammatory chemokines decreases the analgesic function of the opioid receptor and therefore decreases the pain threshold at inflammatory sites. What is also interesting that chemokines can overcome the intrinsic resistance of leukocyte recruitment at the blood–brain barrier (BBB). They suggest this based on the observed increase in chemokine expression in the brain resulting in leukocyte migration and the activation of astrocytes and microglial cells. This is particularly interesting with regard to CIPN since the activation of astrocytes and microglial cells plays a part in the development of CIPN.

#### 3.2.3. Epigenetic Regulation of CCL3 via Macrophages

The following study demonstrates another mechanism that induces CCL3 upregulation in the injured peripheral nerve, showing that epigenetic regulation in macrophages controls chemokine expression, particularly CCL3, and may contribute to sustained neuroinflammation and neuropathic pain.

Kiguchi and colleagues state that epigenetic histone modification within infiltrating immune cells (e.g., macrophages), such as acetylation, methylation, and phosphorylation, can induce prolonged gene expression and the upregulation of CCL3 [81]. The same counts for its receptors. This indicates that chemokine cascades may elicit chronic neuroinflammation following nerve injury. They found that epigenetic histone modification in macrophages induces the upregulation of CCL3 in the injured SCN after PSL and administration of histone acetyltransferase (HAT) inhibitor anacardic acid (ACA) leads to inhibition of the upregulation.

The observation that macrophages infiltration induce CCL3 levels is supported by another study performed by Kiguchi et al. in 2015 [82]. The findings revealed that nAChR agonists inhibit pSTAT3 in inflammatory macrophages. pSTAT3 is the phosphorylated (activated) form of STAT3. Since pSTAT3 participates in the expression of inflammatory molecules, including CCL3 [83,84]. It can be hypothesized that its inhibition leads to suppressing the upregulation of CCL3 and IL-1β resulting in a decrease in neuropathic symptoms.

## 4. Summary

All things considered, upregulation of CCL3 can be associated with the development and maintenance of neuropathic pain. Nerve damage in the PNS, which is often seen as a result of chemotherapeutic treatment, activates microglial cells and astrocytes in the CNS, which then produce CCL3. Epigenetic modifications in the macrophages are also proposed as an inducer of the upregulated CCL3. Either way, the accumulation of CCL3 alternates pain processes via cross-sensitization of TRPV1 and cross-desensitization of opioid receptors. Furthermore, the result of the study performed in 2015 by Zhang [82] revealed that the activation of STAT3 correlates with the upregulation of CCL3 expression. Since STAT3 is considered to be a therapeutic target in colorectal cancer [85], this would be interesting to further investigate. Unfortunately, so far this is the only study that provides evidence for this correlation and there was no study found regarding the association of STAT3 and CCL3 in colon cancer.

### Link Between Colon Cancer and CIPN, with a Focus on CCL3

It is clear that CCL3 is involved in the pathophysiology of colon cancer: most studies conclude that the elevated level of CCL3 is associated with the development and maintenance of colon cancer [17,18,28,31,36,39,42,44,49]. This is induced by its binding to its receptor, CCR5 [31,36,39]. Different mechanisms were proposed; via TRAF6/NF-κB-signaling [17,27], MAPK2 pathway [28] and Erk, PI3K-AKT and Wnt signaling [39,40]. Pathak et al. [40,44] and Pervaiz et al. [40] proposed that the upregulation of CCL3 is associated with the downregulation of TP53 via CCR5-mediated cell cycle-related cascades. Conversely, the studies of Song et al. [50], Allen et al. [52,53] and Yuan et al. [54] found an inverse correlation between the elevated levels of CCL3 and colon cancer. Even though the study of Song et al. found a modest correlation between upregulation of CCL3 and colon cancer, which was not significant after FDR correction. The discrepancies elucidate the complex and intertwined mechanism in which the chemokine is involved, and further research is necessary.

In addition to its role in colon cancer pathophysiology, CCL3 also seems to be involved in neuropathy [57,58,60,64,65,66,67,68,69,71,72,73,77,81,82,86,87]. The exact mechanism by which CCL3 could drive the development of neuropathy (and CIPN within that) is not fully understood. Based on current literature, CCL3 might be activating the MAPK pathway [65,73] leading to the upregulation of pro-inflammatory cytokines and enhanced neuronal excitability, which are both implicated in the development and maintenance of neuropathic pain. In addition, CCL3 seems to be driving alternations of the pain perception. This includes the activation of P2X7R [58] and sensitization of TRPV1 [73,77] that results in a decrease in the pain threshold. CCL3 has also been shown to desensitize opioid receptors involved in analgesia via heterologous crosstalk [80].

Based on all these findings, CCL3 could be a direct link between colon cancer and the development of CIPN, potentially serving as a risk factor for the latter. By looking at the proposed pathways in which CCL3 is involved in colon cancer and neuropathic pain, one is found to directly overlap. Phinney et al. [28] reported that activation of the MAPK pathway in tumor cells and macrophages leads to CCL3 upregulation, which subsequently drives tumor cell proliferation. Their study also revealed that CCL3 can activate the MAPK pathway, resulting in a feedback loop. Sun et al. [65] demonstrated that this pathway is also highly expressed in the spinal microglia and is involved in the chemokine-cytokine mediated regulation of neuropathic pain. They also show that CCL3 induces microglia activation in the spinal cord via the MAPK pathway, which results in neuropathic pain. This mechanism is supported by Kiguchi et al. [73,86]. So, how might CCL3 then induce neuropathy? According to the studies, CCL3 can induce neuropathy by sensitization of TRPV1 [73,77], P2X7R [58], or heterologous desensitization of the opioid receptors [80].

Even more interestingly, CRC could drive the development of CIPN, with CCL3 acting as an important factor. This is supported by the study of De la Fuente Lopez [18], which suggests that elevated plasmatic levels of CCL3 indicate a systemic inflammatory condition in the patient. Based on studies discussed in the CIPN chapter, systemic inflammation is well established as a cause of neuronal damage, which can lead to neuropathic pain according to the mechanisms described above.

The studies of Phinney et al. [28] and Kiguchi et al. [82] also draw a possible link between colon cancer and CIPN, (at least partially) via CCL3. As already elaborated, tumor-cell-derived CCL3 activates the MAPK pathway, leading to the production of IL-1β, IL-6 and TNF-α. Kiguchi et al. shows that the macrophage-induced upregulation of CCL3 is driven by pSTAT3, which is activated by JAK, itself activated by various cytokines including IL-6. It is feasible that this upregulation of CCL3 in the spinal cord can lead to neuropathy or could drive its development. Therefore, the overlapping CCL3–CCR5–MAPK signaling axis may represent a mechanistic link between colorectal cancer progression and the development of chemotherapy-induced peripheral neuropathy. Figure 3 summarizes the proposed mechanisms by which CCL3 may contribute to both pathologies.

However, some of the discrepancies between the outcome of the studies, such as conflicting findings on whether CCL3 levels are increased or decreased in CRC, and the various conditions and models applied make it difficult to suggest a clear link between the pathologies, so the proposed mechanisms stay with speculations. Further investigation is necessary to attain a comprehensive understanding of the complex pathway network.

In addition, we should acknowledge the closely related chemokine CCL3L1, which differs by only a few amino acids but exhibits significantly higher affinity and potency for the CCR5 receptor, a key player in immune regulation and HIV infection [88]. Unlike CCL3, CCL3L1 shows considerable copy number variation in humans, influencing disease susceptibility and immune responses [89,90]. However, the current literature on CCL3L1 in the context of colorectal cancer or chemotherapy-induced peripheral neuropathy is very limited, and no studies have directly compared its role to that of CCL3 in these settings. As such, our focus remains on CCL3, for which most of the available mechanistic, animal, and clinical evidence has been generated.

To date, most mechanistic and animal model studies have concentrated on CCL3, which might not entirely reflect the distinct roles of CCL3L1 in human disease. However, the literature data on CCL3L1 is limited, especially with regard to neuropathy and cancer. Again, further investigation is necessary.

To provide a structured overview of the signaling cascades implicated in both colorectal cancer and chemotherapy-induced peripheral neuropathy, Table 1 summarizes the key pathways, their roles, and the mechanisms proposed.

## 5. Conclusions

Based on current literature data, it is obvious that CCL3 has an important role both in the development of colon cancer and neuropathy. One may hypothesize that colon cancer contributes to neuropathic pain, at least in part through elevated levels of CCL3, which exerts neurotoxic and pro-nociceptive effects. This could suggest that the blood level of CCL3 could be considered a biomarker for the development of CIPN. However, more investigation should be performed regarding this linkage.

Previous studies have demonstrated robust cancer-induced systemic changes, characterized by elevated inflammatory mediators such as IL-6, TNF-α, CXCL10, and MMP9 [91]. CCL3 may be part of this inflammatory network, representing another molecular link between CRC and neuropathic damage.

The current evidence supports a potential role for CCL3 not only as a diagnostic biomarker but also as a possible therapeutic target for CIPN and possible CRC itself. CCL3 and its connection to cancer and CIPN is a fascinating target for future cancer neuroscience studies. We hope that this review lays the groundwork for future research to further elucidate the exact role of CCL3 in these interrelated pathologies.

## Figures and Tables

**Figure 1 biomedicines-13-02512-f001:**
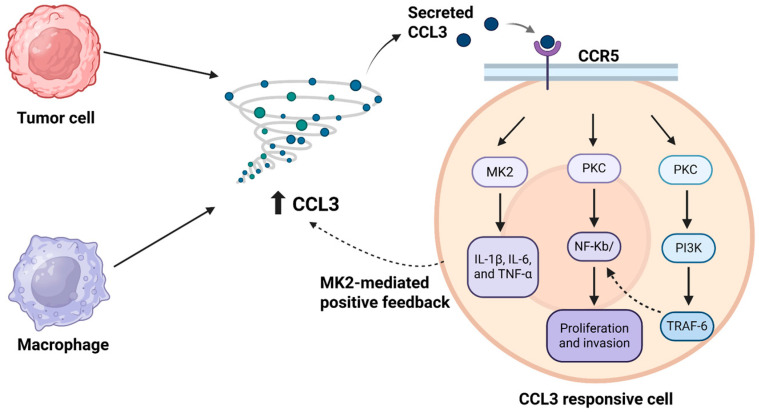
CCL3/CCR5–MAPK signaling drives CRC proliferation and invasion. Tumor cells and infiltrating macrophages secrete elevated levels of CCL3, which binds CCR5 on a generalized MAPK-activated target cell. CCR5 engagement splits into three downstream arms: (1) MK2 activation induces IL-1β, IL-6 and TNF-α release and feeds back to further up-regulate CCL3; (2) PKC → NF-κB promotes inflammation and tumor invasion; and (3) PKC → PI3K → TRAF-6 amplifies chemokine production. Created in BioRender. Rosa, C. (2025) https://BioRender.com/pgssr1u.

**Figure 2 biomedicines-13-02512-f002:**
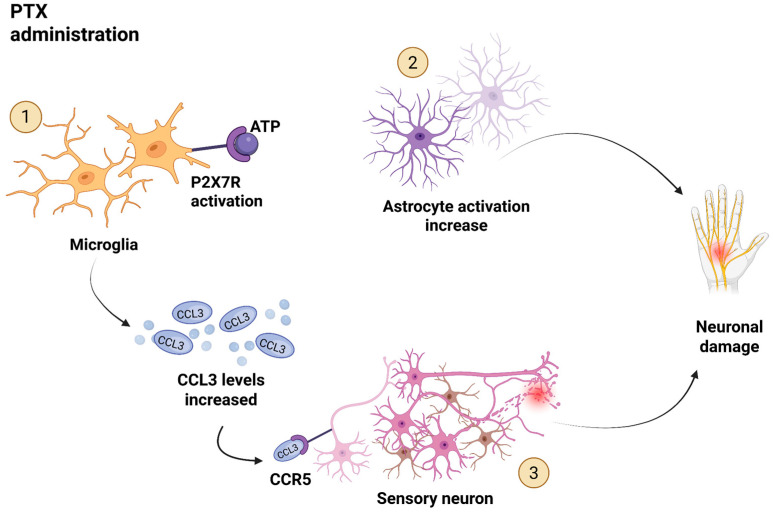
Proposed mechanism of CCL3-driven neuroimmune alterations in paclitaxel (PTX)-induced CIPN. PTX administration (top left) leads to activation of spinal microglia, marked by P2X7R stimulation through extracellular ATP (1). This activation promotes the release of CCL3, which in turn binds to CCR5 receptors on sensory neurons (3), contributing to neuronal injury and mechanical allodynia. Concurrently, PTX also induces astrocyte activation in the spinal cord (2), which has been associated with neuronal damage. Created in BioRender. Rosa, C. (2025) https://BioRender.com/c4ou27a.

**Figure 3 biomedicines-13-02512-f003:**
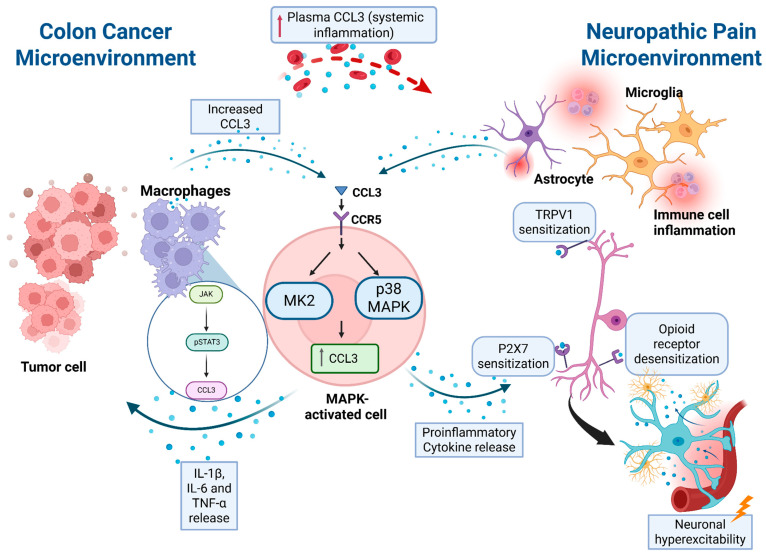
CCL3-driven MAPK signaling as a shared mechanism in colon cancer and neuropathic pain microenvironments. In the colon cancer microenvironment (left), tumor cells and macrophages secrete CCL3, which binds to CCR5 on a MAPK-activated cell, leading primarily to MK2 activation. This triggers additional CCL3 and pro-inflammatory cytokine (IL-1β, IL-6, TNF-α) release, forming a feedforward loop that sustains local inflammation and may contribute to tumor progression. The red dotted arrow at the top represents systemic inflammation driven by elevated plasma CCL3, which may bridge to the neuropathic pain microenvironment (right). There, injured sensory neurons and glial cells release CCL3, which binds CCR5 on microglia, activating p38 MAPK and driving TRPV1 and P2X7 sensitization and opioid receptor desensitization. Solid arrows indicate causal signaling or cytokine diffusion. Created in BioRender. Rosa, C. (2025) https://BioRender.com/46kxrtm.

**Table 1 biomedicines-13-02512-t001:** Summary of signaling pathways involving CCL3 in colorectal cancer (CRC) and neuropathy, including their proposed mechanisms.

Pathway	Role in CRC	Role in Neuropathy	Proposed Mechanism
TRAF6/NF-κB	Drives tumor development through TRAF6/NF-κB signaling.	Not directly implicated.	Upregulation of CCL3 via NF-κB contributes to tumor proliferation, with cross-talk to MAPK.
MAPK	Promotes tumor cell proliferation when activated in tumor cells and macrophages.	Activates microglia and astrocytes, leading to pro-inflammatory cytokine release and neuronal excitability.	CCL3 activates MAPK, creating a feedback loop that amplifies inflammation and pain sensitization
PI3K–AKT	Activated in CRC cells following chemokine stimulation.	Upregulated in models involving p53-dependent regulation of CCL3.	Enhances proliferation and survival signaling; links to miRNA regulation of CCL3 and VEGF.
Wnt	Identified as one of the pathways associated with CCL3 upregulation.	Found in conjunction with PI3K–AKT activation in neuropathic models.	Supports tumorigenesis and possibly influences CCL3 expression through p53–miRNA regulation.
p53	Downregulation of p53 associated with increased CCL3.	CCL3 upregulation dependent on p53 status; absent in p53-null cells.	p53 status regulates CCL3 and VEGF expression, linking tumor stress responses to chemokine levels.
STAT3	Considered a therapeutic target in CRC.	pSTAT3 in macrophages drives CCL3 and IL-1β expression, contributing to neuropathic pain	JAK/STAT3 activation upregulates CCL3; inhibition suppresses inflammation and pain.
TRPV1	Not implicated in CRC.	Sensitization of nociceptive neurons via CCL3–CCR1 signaling increases TRPV1 activity.	CCL3 activates PLC–PKC pathway, phosphorylating TRPV1 and enhancing thermal hyperalgesia.
P2X7R	Not implicated in CRC.	Microglial P2X7R activation promotes CCL3 release and sustains CIPN.	P2X7R–CCL3 circuit amplifies microglial activation, lowering pain thresholds.
Opioid receptor desensitization	Not implicated in CRC.	Chemokines desensitize opioid receptors, reducing analgesic function	CCR1/CCL3 (and related chemokines) cross-desensitize opioid GPCRs, lowering pain thresholds.

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
