# Peer review of "Unveiling the Role of CCL3: A Driver of CIPN in Colon Cancer Patients?"

_biomedicines, 2025, doi:10.3390/biomedicines13102512_

Round 1
Reviewer 1 Report
Comments and Suggestions for Authors
- Line 18 - Specifically, we evaluate whether CCL3 may serve as a molecular link between cancer progression and the development of neuropathic pain. Did you evaluate any experimental assay?
- Line 44 - Studies have shown that a large number of patients that undergo chemotherapy (still a cornerstone of CRC treatment), have to deal with terrible side effects. Reference needs to be updated.
- Chemotherapy induces apoptosis, mitochondrial damage and increases in reactive oxygen species (ROS), triggering various molecular pathways that upregulate pro-inflammatory cytokines and chemokine – Reference needs to be update.
- Line 214 - DDS is used to mimic the pathologic changes of human ulcerative colitis – its DSS or DDS?
- Line 228 - This revealed that heparin-binding epidermal growth factor (HB-EGF) expression was depressed in CCL3/CCR5-deficient mice. Double-colour immunofluorescence showed that HB-EGF was expressed by type I collagen-positive cells. – Did you attach any images for these results? Sentence needs to be modified.
- Line 240 - concludes that CCL3-CCR5-mediated fibroblasts – Sentence needs to be checked.
- Line 241 - Similar roles for fibroblast-derived signaling in tumor growth have also been demonstrated in models of colitis-associated neoplasia. – What models? Can you please explain and include in the manuscript.
- Tanabe with Sasaki et al. – its Tanabe et al or Sasaki et al?
- Line 285 - Nishikawa proposed a different point of view on tumor development focusing on the bone marrow (BM) – Should be changed to Nishikawa et al.
- Line 323 - (nucleoside analog, chemotherapeutic agent [40]) – Reference should be cited to the end of the statement.
- Line 326 - Furthermore, they found that CCR5 targeting suppresses colony formation and the migration of CRC cells. – Who found, it should include in the manuscript.
- The researchers hypothesize that exposure of these GPCRs to their agonist – GPCRs full form should be reported in first time.
- Abbreviations should be included in the manuscript.
- Funding source, Acknowledgement, conflict of interest, Authors contribution should be included in the manuscript.
- If possible should be include tables, its very easy to understand the mechanisms of signaling pathways.
Author Response
We sincerely thank the Reviewer for the careful reading of our manuscript and for the thoughtful and constructive feedback. We greatly appreciate the insightful comments, which have helped us improve the clarity, accuracy, and overall quality of our manuscript. Below, we provide a detailed response to each reviewer’s suggestions. All changes in the revised manuscript are highlighted
Comment 1 (Line 18): Did you evaluate any experimental assay?
Response: Thank you for pointing this out. We clarified in the revised manuscript that this review is based on preclinical and clinical studies reported in the literature rather than on experiments performed by the authors. The sentence was rephrased accordingly (p.2, Abstract).
Comment 2 (Line 44): Reference needs to be updated regarding chemotherapy side effects.
Response: The outdated reference has been replaced with more recent literature on chemotherapy side effects (p.3).
Comment 3: Reference update for chemotherapy-induced apoptosis, mitochondrial damage, ROS, and cytokine upregulation.
Response: We have updated this statement with current references (p.4).
Comment 4 (Line 214): DDS or DSS?
Response: Corrected to DSS (dextran sulfate sodium) .
Comment 5 (Line 228): Sentence modification regarding HB-EGF images.
Response: The sentence was rephrased to describe the findings from Sasaki et al. without implying that images are included in our review
Comment 6 (Line 240): Sentence needs to be checked.
Response: Revised for clarity
Comment 7 (Line 241): Please explain the models used.
Response: Expanded to specify that the models included an AOM/DSS-induced colitis-associated CRC model and a lung metastasis model
Comment 8: Tanabe with Sasaki et al.—which is correct?
Response: Corrected to “Tanabe et al.” where appropriate
Comment 9 (Line 285): Should be “Nishikawa et al.”
Response: Corrected to “Nishikawa et al.”
Comment 10 (Line 323): Reference should be cited at the end of the statement.
Response: Reference repositioned to the end of the sentence
Comment 11 (Line 326): Please specify who found that CCR5 targeting suppresses colony formation and migration.
Response: Sentence revised to specify that these findings were reported by Pervaiz et al.
Comment 12: GPCRs full form should be reported at first mention.
Response: Expanded to “G protein–coupled receptors (GPCRs)” at first mention
Comment 13: Abbreviations should be included.
Response: A comprehensive list of abbreviations has been added at the end of the manuscript.
Comment 14: Funding source, Acknowledgements, Conflict of interest, and Author contributions should be included.
Response: These sections have been filled according to the Guidelines for Authors.
Comment 15: Tables should be included to better summarize pathways.
Response: A new summary table has been added (Table X) to illustrate key signaling pathways and mechanisms.

Reviewer 2 Report
Comments and Suggestions for Authors
Recommendation
This manuscript is a valuable contribution to the understanding of the intersection between colorectal cancer biology and chemotherapy-induced neuropathy. The minor revisions suggested will enhance clarity, strengthen clinical relevance, and improve overall presentation.
Accept with minor revisions.
Specific Suggestions
- Emphasize the dual role of CCL3 in CRC and CIPN earlier in the manuscript
- Expand the section on potential clinical applications (biomarkers, therapeutic targeting) in detail along with latest case studies.
- Provide a more detailed analysis of discrepant findings in clinical studies.
- Discuss the the significance of CCL3L1 and its variation.
- Improve the figure clarity and visibility.
- Proofread the manuscript carefully for minor grammatical and formatting issues.
Author Response
We thank the reviewer for taking the time to carefully read our manuscript. We are grateful for the positive assessment and for suggesting ways to strengthen the clinical relevance of the manuscript.
-
Emphasize the dual role of CCL3 earlier.
→ The Introduction now highlights the dual role of CCL3 in both CRC and CIPN -
Expand clinical applications (biomarkers, therapeutic targeting).
→ We did our best effort to include and expand upon (possible) clinical applications throughout the manuscript. All changes are marked red. -
Provide detailed analysis of discrepant findings.
→ We expanded the discussion to address discrepancies and possible explanations (e.g., on p. 10). -
Discuss the significance of CCL3L1 variation.
→ We expanded the discussion on the relevance of CCL3L1 variation has been added (p. 17). -
Improve figure clarity and visibility.
→ Figures have been revised for clarity, and legends have been expanded -
Proofread for grammatical/formatting issues.
→ The manuscript has been carefully proofread by a native speaker and formatting errors corrected.
Reviewer 3 Report
Comments and Suggestions for Authors
In this manuscript, Luzac and coauthors provide an updated overview of the role of the chemokine CCL3 in the pathogenesis of colorectal cancer (CRC) and chemotherapy-induced peripheral neuropathy (CIPN). The authors first summarize studies investigating the role of CCL3 in tumorigenesis, highlighting its conflicting functionality—many studies report a pro-tumorigenic role, while others demonstrate tumor-suppressive effects. They then review studies on CCL3 in CIPN, showing its direct involvement in promoting neural inflammation and neuropathy. Based on these findings, the authors propose CCL3 as a potential mechanistic link between CRC progression and CIPN, which is both novel and clinically relevant. Overall, the manuscript is well written, comprehensive, and supported by wide literature coverage.
Despite the valuable discussion of CCL3 in CRC and CIPN, the manuscript lacks logical clarity in its presentation and contains significant redundancy in multiple sections. To improve clarity and readability, it is strongly recommended to streamline the structure, remove unnecessary titles/sections, and reduce repeated discussion of the same studies.
The following comments may help improve the manuscript:
- After Section 2 (summarizing the positive correlation of CCL3 with tumor growth), the authors move to Section 3 (p53 involvement in CCL3 regulation), Section 4 (chemokine–chemokine receptor axis as a treatment option), and Section 5 (negative correlation of CCL3 with tumor growth). Arrangement of these sections indicates a lack of a clear logical flow. A similar issue is seen in Sections 7–11. It is recommended to combine certain sections for better coherence. For example, Sections 3–4, 6, and 9–11 could be merged into other sections to streamline the narrative and improve logical continuity.
- There is substantial redundancy in the literature summary and discussion. For instance, lines 191–204 largely repeat material already presented in Sections 2 and have high redundancy to Sections 6 and 12. The redundancy makes the manuscript unnecessarily long and less readable. To enhance readability, unnecessary summaries should be removed and repeated mentions of the same studies across different sections should be minimized.
- Figure 3 essentially reproduces the combined content of Figures 1 and 2. It is recommended to remove Figures 1 and 2 to reduce redundancy.
- All abbreviations should be spelled out in full at first mention in the manuscript. Some abbreviations are introduced in the manuscript without definition. In addition, all abbreviations used in the figures and figure legends should be defined in the legends.
- Numerous grammatical errors are present throughout the text. Careful proofreading or professional English editing is recommended to improve overall readability.
Author Response
We sincerely thank the Reviewer for the careful reading of our manuscript and for the thoughtful and constructive feedback. We greatly appreciate the insightful comments, which have helped us improve the clarity, accuracy, and overall quality of our manuscript. Below, we provide a detailed response to each reviewer’s suggestions. All changes in the revised manuscript are highlighted.
-
Logical flow and section arrangement.
→ We reorganized the manuscript and the titles of the different chapters. The manuscript has been thoroughly revised, with particular attention to language and flow. -
Redundancy in literature summaries.
→ Redundant text has been condensed throughout the manuscript. With the help of a native English speaker, we have revised the whole text of the manuscript to enhance readability and to correct mistakes. All new additions and rewritten parts are marked in the text. -
Figures (redundancy between Fig. 3 and Figs. 1–2).
→ Figure 2 was expanded to include additional neuronal damage mechanisms and to reduce overlap, figures have also been re-worked to elevate clarity. -
Abbreviations.
→ All abbreviations are now spelled out at first mention and defined in legends. Also, in accordance with a comment from another reviewer, an abbreviations list has been added. -
Grammar and readability.
→ The manuscript has undergone extensive language editing by a native speaker to correct grammar and improve readability.
Reviewer 4 Report
Comments and Suggestions for Authors
Overall, the quality of the manuscript is excellent. This literature review is well-written and accessible, with a compelling topic that warrants further exploration.
While some aspects of CCL3 activity in CRC and chemotherapy-associated CIPN are already recognized—such as its involvement in modulating tumor inflammation and the correlation between altered CCL3 levels and tumor progression, invasiveness, and metastasis—the precise role of CCL3 within CRC remains complex and likely varies according to individual immune and molecular contexts. This underscores the need for continued investigation into its specific functions.
Furthermore, preliminary studies hint that fluctuations in chemokine levels like CCL3 could be associated with the severity of neuropathy, although this area remains in early stages and requires substantial expansion.
The authors have effectively compiled existing research findings.
In the conclusion, it would be useful to improve the discussion by elaborating on the emerging insights from cancer neuroscience research that explore the connection between CCL3, tumor biology, and neuropathic pain mechanisms. and emphasized the importance of further studies to clarify these relationships.
However, some corrections are needed to improve the quality of the manuscript. I have a few minor comments for the authors:
-improve figures and enhance legends by providing more detailed descriptions.
-some formatting errors.
Author Response
We thank the Reviewer for their careful reading, positive evaluation, and constructive feedback. Their comments greatly helped us improve the clarity and quality of the manuscript. Below, we provide detailed responses to each suggestion. All changes are highlighted in the revised version.
-
Conclusion – emerging insights in cancer neuroscience.
→ The conclusion has been expanded to discuss recent advances in cancer neuroscience, emphasizing links between CCL3, tumor biology, and neuropathic pain. -
Figures and legends.
→ Figures have been improved, and legends enhanced with more detailed explanations. -
Formatting errors.
→ Formatting errors have been corrected throughout the manuscript. The entire manuscript has been carefully revised by a native English speaker with expertise in scientific writing, and several parts of the text were reworded for improved clarity.
Round 2
Reviewer 3 Report
Comments and Suggestions for Authors
In the revised manuscript, Luzac and coauthors have addressed most of the comments from the initial review, which has greatly improved the logical flow and reduced redundancy. All abbreviations are now clearly defined, and the overall language has been noticeably refined. Therefore, I support acceptance and publication of the manuscript.